



# DINCAE 2: multivariate convolutional neural network with error estimates to reconstruct sea surface temperature satellite and altimetry observations

Alexander Barth[1], Aida Alvera-Azcárate[1], Charles Troupin[1], and Jean-Marie Beckers[1]

[1]GHER, University of Liège, Liège, Belgium

**Correspondence:** A. Barth (a.barth@uliege.be)

**Abstract.** DINCAE (Data INterpolating Convolutional Auto-Encoder) is a neural network to reconstruct missing data (e.g. obscured by clouds or gaps between tracks) in satellite data. Contrary to standard image reconstruction (in-painting) with neural networks, this application requires a method to handle missing data (or data with variable accuracy) already in the training phase. Instead of using a standard L2 (or L1) cost function, the neural network (U-Net type of network) is optimized

by minimizing the negative log likelihood assuming a Gaussian distribution (characterized by a mean and a variance). As a consequence, the neural network also provides an expected error variance of the reconstructed field (per pixel and per time instance).

In this updated version DINCAE 2.0, the code was rewritten in Julia and a new type of skip connection has been implemented which showed superior performance with respect to the previous version. The method has also been extended to handle

multivariate data (an example will be shown with sea-surface temperature, chlorophyll concentration and wind fields). The improvement of this network is demonstrated in the Adriatic Sea.

Convolutional networks work usually with gridded data as input. This is however a limitation for some data types used in oceanography and in Earth Sciences in general, where observations are often irregularly sampled. The first layer of the neural network and the cost function have been modified so that unstructured data can also be used as inputs to obtain gridded fields

as output. To demonstrate this, the neural network is applied to along-track altimetry data in the Mediterranean Sea. Results from a 20-year reconstruction are presented and validated. Hyperparameters are determined using Bayesian optimization and minimizing the error relative to a development dataset.

## 1 Introduction

Ocean data are generally sparse and inhomogeneously distributed. The data coverage often contains large gaps in space and

time. This is in particular the case of in situ observations. Satellite remote sensing only measures the surface of the ocean but generally has better spatial coverage than in situ observations. But still about 75% of the ocean surface is on average covered by clouds that block sensors in the optical and infrared bands (Wylie et al., 2005). Given the sparsity of data, it is natural to aim to combine data representing different parameters as e.g. mesoscale flow structures are often visible in all ocean tracers.





Prior work on using multivariate data in connection with satellite data use for example empirical orthogonal functions (EOF) who can be naturally extended to multivariate datasets as long as an appropriate norm is defined. For example, Alvera-Azcárate et al. (2007) uses sea surface temperature, chlorophyll, and wind satellite fields with Data Interpolating Empirical Orthogonal Functions (DINEOF). Multivariate EOFs have also been used to project surface observations to deepers layers Nardelli and

Santoleri (2005) or to derive nitrate maps in the Southern Ocean (Liang et al., 2018). In the latter case, EOFs linking salinity, and potential temperature and nitrate concentration are derived from model simulations.

As some observations can be measured at much high spatial resolution via remote sensing (in particular the resolution of sea-surface temperature is much higher than the resolution of sea surface salinity products), the so-called "multifractal fusion techniques" are used to improve remote sensed surface salinity estimates using sea surface temperature. The data fusion is

implemented as a locally weighted linear regression (Olmedo et al., 2018, 2021). Han et al. (2020) also used an earlier version of the DINCAE code to estimate sea-surface chlorophyll using additional sea surface temperature observations.

The structure of a neural network and in particular its depth is uncertain and to some degree dependent on the used data set. We also investigate the use of deeper neural networks in this work. It is known that deeper neural networks are more difficult to train because of the well-known vanishing gradient problem (Hochreiter, 1998): the derivative of the loss function relative to

the weights of the first layers of a neural network have the tendency to either decrease (or increase) exponentially with a increasing number of layers. This prevents effective optimization (training) of these layers using gradient-based optimization methods.

Several methods have been proposed in the literature to mitigate such problems using alternative neural network architectures. In the context of the present manuscript, skip connections in the form of residual layers have been tested (similar to

residual networks, He et al., 2015). The derivative of the loss function relative to the weights in such layers remains (at least initially) closer to one, so there is a more direct relationship between the loss function and the weights and biases to be optimized. Deeper residual networks include shallower networks as a special case and there, per construction, should perform at least as good as shallower networks.

Gradient of a whole network is computed via back-propagation which is essentially based on the repeated application of the chain rule for differentiation. The information of the observation is injected via the loss function and propagated backward in a way which is similar to the 4D-var backward in time integration of the adjoint model. Another interesting neural network architecture has been proposed in the form the Inception network Szegedy et al. (2015) where the output of intermediate layers, here in the form of a preliminary reconstruction, are used in the loss function (in addition to the output of the final layer). The

result is that the information of the observations are injected not only at the final layer but also in the intermediate layer which also contributes to reducing the vanishing gradient problem.

While for gridded satellite data, approaches based on empirical orthogonal functions and convolutional neural networks have been shown successful, it is difficult to apply similar concepts to non-gridded data as these methods typically require a





stationary grid. Another objective of this paper is to show how convolutional neural networks can be used on non-gridded data. This approach is illustrated with altimetry observations.

The objective of this manuscript is to highlight the improvement of DINCAE made relative to the previously published version Barth et al. (2020). The section 2 presents the updated structure of the neural network. The gridded and non-gridded observations used here are presented in section 3. Details of the implementation are also given (section 4). The results and conclusions are presented in sections 5 and 6.

## 2 The neural network architecture

The DINCAE network (Barth et al., 2020) is a neural network composed of an encoder and decoder network. The encoder uses the original gappy satellite data (with additional metadata as explained later) and the decoder uses the output of the encoder to reconstruct the full data image (along with an error estimate). The encoder uses a series of convolutional layers followed by max pooling layers reducing the resolution of the datasets. The decoder does essentially the reverse operation by using convolutional and interpolation layers. This is the general structure of a convolutional autoencoder and the classical U-Net networks (Ronneberger et al., 2015). In the following section we will discuss the main components of the DINCAE neural network used for the different test cases and emphasize the changes relative to the previous version.

### 2.1 Skip connections

In an autoencoder, the inputs are compressed by forcing the data flow through a bottleneck which ensures that the neural network must efficiently compress and decompress the information. However, in U-Net (Ronneberger et al., 2015) and DINCAE skip connections are implemented allowing the information flow of the network to partially bypass the bottleneck to prevent the loss of small scale details in the reconstruction. Skip connections can be realised either by concatenating tensors along the feature map dimension (as it is done in U-Net) or by summing the tensors.

$$\mathbf{X}^{(l+1)} = \mathrm{cat}(f^{(l)}(\mathbf{X}^{(l)}), \mathbf{X}^{(l)}) \tag{1}$$

where $\mathrm{cat}$ concatenates two 3-dimensional arrays along the dimension representing the features channels. Sum skip-connections are implemented as:

$$\mathbf{X}^{(l+1)} = f^{(l)}(\mathbf{X}^{(l)}) + \mathbf{X}^{(l)} \tag{2}$$

Clearly, the output of a cat-skip connection has a size twice as large as the output of a sum-skip connection. These skip connections are followed by a convolutional layer, which ensures that the number of output features are the same for both types of skip connection. In fact, one can show that the sum-skip connection (followed by a convolution layer) is formally a special





case of the cat-skip connection. However, sum-skip connections can have an advantage because the weight and basis of the convolutional layers are more directly related to the output of the neural network which helps to reduce the so-called vanishing gradient problem (He et al., 2015).

## 2.2 Refinement step

The whole neural network can be described as two functions that provide the input variable product between the reconstruction $\hat{y}$ and its expected error error variance $\hat{\sigma}$ for every grid cell. The loss function is derived from the negative log-likelihood of a Gaussian with mean $\hat{y}$ and standard deviation of $\sigma$:

$$J(\hat{y}_{ij}, \hat{\sigma}_{ij}) = \frac{1}{2N} \sum_{ij} \left[ \left( \frac{y_{ij} - \hat{y}_{ij}}{\hat{\sigma}_{ij}} \right)^2 + \log(\hat{\sigma}_{ij}^2)) \right] \tag{3}$$

For a convolutional auto-encoder with refinement, the intermediate outputs $\hat{y}$ and $\hat{\sigma}$ are concatenated with the inputs and
passed through another auto-encoder with the same structure (expect for the number of filters for the input layer which has to accommodate the two additional fields corresponding to $\hat{y}$ and $\hat{\sigma}$). The weights of the first and second auto-encoder are not related. The final cost function with refinement $J_r$ is given by:

$$J_r = \alpha J(\hat{y}_{ij}, \hat{\sigma}_{ij}) + \alpha' J(\hat{y}'_{ij}, \hat{\sigma}'_{ij}) \tag{4}$$

where $\hat{y}'$ and $\hat{\sigma}'$ are the reconstruction and its expected error variance produced by the second auto-encoder. The weights $\alpha$
and $\alpha'$ control how much importance is given to the intermediate output (relative to the final output).

With a refinement step, the neural network becomes essentially twice as deep and the number of parameters (approximately) doubles. The increased depth would make it prone to the vanishing gradient problem. However, by including the intermediate results in the cost function this problem is reduced.
The refinement step has been used in image in-painting for computer vision application (Liu et al., 2019) and it has also been applied for oceanographic data for tide gage data (Zhang et al., 2020). In the present work only one refinement step is tested, but the code supports an arbitrary number of sequential refinement steps.

## 2.3 Multivariate reconstructions

Auxiliary satellite data (with potentially missing data) can be provided for the reconstruction. The handling of missing data
in these auxiliary data is identical to the way missing data are treated for the primary variable. For every auxiliary satellite data, the average over time is first removed. The auxiliary data (divided by its corresponding error variance) and the inverse of



the error variance are provided as input. Where data is missing, the corresponding input values are set to zero representing an infinitely large error. Multiple time instances centred around a target time can be provided as input.

## 2.4 Non-gridded input data

Current satellite altimetry mission measures sea surface height along the ground track of the satellite. Satellite altimetry can
measure through clouds but the data is only available along a collection of tracks. In order to handle better such data sets, we extended DINCAE to handle unstructured data as input.

The first layer in DINCAE is a convolutional layer which typically requires a field discretized on a rectangular grid. The convolutional layer can be seen as the discretized version of the following integral:

$$g(x,y) = \int_{\Omega_w} w(x'-x, y'-y) f(x,y) dx' dy' \tag{5}$$

where $f$ is the input field, $w$ are the weights in the convolution (also called convolution kernel) , $\Omega_w$ is the support of the function $w$ (i.e. the domain where $w$ is different from zero) and $g$ the output of the convolutional layer. To discretize the integral, the continuous function $f$ is replaced by a sum of Dirac functions using the values $f_{i,j}$ defined on a regular grid:

$$f(x,y) = \sum_{i,j} f_{i,j} \delta(x - i\Delta x, y - j\Delta y)$$

In this case, the continuous convolution becomes the standard discrete convolution as used in neural networks. The weights $w(x,y)$ need only to be known at the discrete locations defined by the underlying grid.
For data points which are not defined on a regular grid we essentially use a similar approach. The function $f$ is again written as a sum of Dirac functions:

$$f(x,y) = \sum_{k} f_k \delta(x - x_k, y - y_k)$$

where now $f_k$ are the values at the locations $(x_k, y_k)$ which can be arbitrary. In order to evaluate the integral (5), it is necessary to know the weights at the location $(x_k, y_k)$. The weights are still discretized on a regular grid, but are interpolated bilinearly to the data location to evaluate the integral. In fact, instead of interpolating the weights $w$, one can also apply the adjoint of
the linear interpolation to $f_k$ (which is mathematically equivalent). This has the benefit that the computation of the convolution can be implemented using the optimized functions in the neural network library.

For data defined on a regular grid, it has been verified numerically that this proposed approach and the traditional approach to compute the convolution give the same results.





## 3 Data

The improvements are examined on two test cases. For multivariate gridded data the approach is tested with sea surface temperature, chlorophyll and winds on the Adriatic Sea, and for non-gridded data altimetry observations on the whole Mediterranean Sea was used. As the altimetry observations do not resolve as many small-scales as sea surface temperature, a larger domain

was chosen for the altimetry test case.

### 3.1  Gridded data (Adriatic Sea)

As the previous application (Barth et al., 2020) considered relatively low resolution AVHRR data, we used more modern and higher resolution satellite data in this application for the Adriatic Sea. The datasets used include:

– Sea Surface Temperature (MODIS Terra Level 3 SST Thermal IR Daily 4km Nighttime v2014.0, doi:10.5067/MODST-

1D4N4) made available by PO.DAAC (https://podaac.jpl.nasa.gov/, JPL, NASA, US)

– Wind speed (Cross-Calibrated Multi-Platform (CCMP) gridded surface vector winds) made available from Remote Sensing Systems (http://www.remss.com/measurements/ccmp/). These datasets are described in Atlas et al. (2011); Mears et al. (2019); Wentz et al. (2019). This dataset has a 6 h temporal resolution and a 1/4 degree spatial resolution resolution. The wind fields are averaged as daily mean fields.

– Chlorophyll-a from Ocean Biology Processing Group, NASA (US) https://oceancolor.gsfc.nasa.gov/data/overview/ at a 4 km resolution and L3 processing level.

The data sets span the time period 2003-01-01 to 2016-12-31. They are all interpolated (using bi-linear interpolation) on the common grid defined by the SST fields.

As ocean mixing reacts to the averaged effect of the wind speed (norm of the wind vector), we also smoothed the speed with

a Laplacian filter using a time period of 2.2 days and a lag of 4 days (wind speed preceding SST). The optimal lag and time period were obtained by maximizing the correlation between the smoothed wind field and SST from the training data.

### 3.2  Non-gridded data (Mediterranean Sea)

Altimetry data from 1993-01-01 to 2019-05-13 between $7^oW$ and $37^oE$ and $30^oN$ to $46^oN$ from 22 satellite missions operating during this time frame is used. This domain contains essentially the Mediterranean Sea but also a small part of the Bay of

Biscay and the Black Sea. In preliminary studies we found that including the data from adjacent seas can help a neural network to better generalize and to prevent overfitting. The data (SEALEVEL_EUR_PHY_L3_REP_OBSERVATIONS_008_061) is made available by the Copernicus Marine Environment Monitoring Service (CMEMS).

This data was split along the following fractions:

– 70% training data,

– 20% development data,





   – 10% test data.

To reduce the correlation between the different datasets, satellite tracks are not splitted and belong entirely to one of these 3 datasets.

Some experiments of the reconstructed altimetry use gridded sea surface temperature satellite observations as an auxiliary datasets for multivariate reconstruction. We use the AVHRR_OI-NCEI-L4-GLOB-v2.0 datasets (Reynolds et al., 2007; National Centers for Environmental Information, 2016) because it is a single consistent dataset covering the full time period of the altimetry data and because it matches approximately the alimetry dataset in terms of resolved spatial scales.

## 4  Implementation

Python code was first ported from TensorFlow 1.12 to 1.15 reducing the training time from 4.5 to 3.5 hours using a GeForce GTX 1080 GPU and Intel Core i7-7700 CPU. We also considered porting DINCAE to TensorFlow 2. TensorFlow 2 programming interface is however quite different from previous versions. As our group gained familiarity with the Julia programming language (Bezanson et al., 2017), we decided to rewrite DINCAE in Julia. Porting DINCAE to Julia with the package Knet (Yuret, 2016) cut down the runtime from 3.5 to 1.9 hours (thanks to more efficient data transformation) using the AVHRR dataset described in Barth et al. (2020), and using a concatenation skip connection in both cases.

For the Adriatic test case, the input is a 3D array with the dimension corresponding to the longitude, latitude and various parameters (an array of the size 168 x 144 x 10). The input is processed by the encoder which is composed by 5 convolutional layers (with 16, 30, 58, 110 and 209 output filters) with a kernel size of 3 by 3 and a rectified linear unit (RELU) activation function followed by a max-pooling layer with a size of 2 by 2. The RELU activation function is commonly used in neural networks and is defined by $f(x) = \max(x, 0)$.

The output of the encoder is transformed back to a full image by the decoder which mirrors the structure of the encoder. The decoder is composed by 5 upsampling layers (nearest-neighborhood interpolation or bilinear interpolation) followed by a convolutional layer with the equivalent number output filters from the encoder (except for the final layer, which has only two outputs related to the reconstruction and its standard deviation). The final layer produces a 3D-array $T^{\text{out}}$ (size 168 x 144 x 2) from which the reconstruction $\hat{y}_{ij}$ and its standard deviation $\hat{\sigma}_{ij}^2$ are computed by:

$$\hat{\sigma}_{ij}^2 \quad = \quad \frac{1}{\max(\exp(\min(T_{ij1}^{(\text{out})}, \gamma)), \mu)} \tag{6}$$

$$\hat{y}_{ij} \quad = \quad T_{ij2}^{(\text{out})} \hat{\sigma}_{ij}^2 \tag{7}$$

where $\gamma = 10$ and $\mu = 10^{-3}$. The min and max functions are introduced here to prevent a division by a value close to zero or the exponentiation of a too large value. This stabilizes the neural network during the initial phase of the training as the weights





| number | type | output size | parameters of the layer |
|---|---|---|---|
| 1 | input | 168 x 144 x 10 | |
| 2 | conv. 2d | 168 x 114 x 16 | n. filters = 16, kernel size = (3,3) |
| 3 | pooling 2d | 84 x 72 x 16 | pool size = (2,2), strides = (2,2) |
| 4 | conv. 2d | 84 x 72 x 30 | n. filters = 30, kernel size = (3,3) |
| 5 | pooling 2d | 42 x 36 x 30 | pool size = (2,2), strides = (2,2) |
| 6 | conv. 2d | 42 x 36 x 58 | n. filters = 58, kernel size = (3,3) |
| 7 | pooling 2d | 21 x 18 x 58 | pool size = (2,2), strides = (2,2) |
| 8 | conv. 2d | 21 x 18 x 110 | n. filters = 110, kernel size = (3,3) |
| 9 | pooling 2d | 11 x 9 x 110 | pool size = (2,2), strides = (2,2) |
| 10 | conv. 2d | 11 x 9 x 209 | n. filters = 209, kernel size = (3,3) |
| 11 | pooling 2d | 6 x 5 x 209 | pool size = (2,2), strides = (2,2) |
| 12 | interpolation | 11 x 9 x 209 | |
| 13 | conv. 2d | 11 x 9 x 110 | n. filters = 110, kernel size = (3,3) |
| 14 | sum output of 13 and 9 | 11 x 9 x 110 | |
| 15 | interpolation | 21 x 18 x 110 | |
| 16 | conv. 2d | 21 x 18 x 58 | n. filters = 58, kernel size = (3,3) |
| 17 | sum output of 16 and 9 | 21 x 18 x 58 | |
| 18 | interpolation | 42 x 36 x 58 | |
| 19 | conv. 2d | 42 x 36 x 30 | n. filters = 30, kernel size = (3,3) |
| 20 | sum output of 19 and 5 | 42 x 36 x 30 | |
| 21 | interpolation | 84 x 72 x 30 | |
| 22 | conv. 2d | 84 x 72 x 16 | n. filters = 16, kernel size = (3,3) |
| 23 | sum output of 22 and 3 | 84 x 72 x 16 | |
| 24 | interpolation | 168 x 144 x 16 | |
| 25 | conv. 2d | 168 x 144 x 2 | n. filters = 2, kernel size = (3,3) |

**Table 1.** Layers of the neural network for the gridded datasets. Note that every convolution is followed by a RELU activation function

.

and biases are randomly initialized.

In Barth et al. (2020), after the convolutional layers the model included two fully-connected layers (with drop-out). This is no longer used as such layers require that the input matrix for training has exactly the same size as the input matrix of 5 the reconstruction (inference) which makes this architecture difficult for large input arrays (which would arise for global or basin-wide sea surface temperature fields for example). The benefit of replacing dense layers by convolutional layers is further discussed in Long et al. (2015).



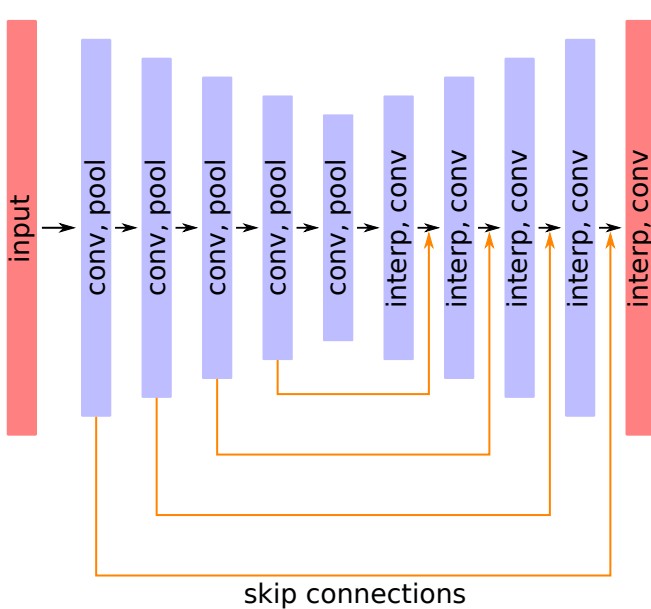

**Figure 1.** General structure of the DINCAE with 2D convolution (conv), max pooling (pool), and interpolation layers (interp). All 2D convolutions are followed by a RELU activation function.





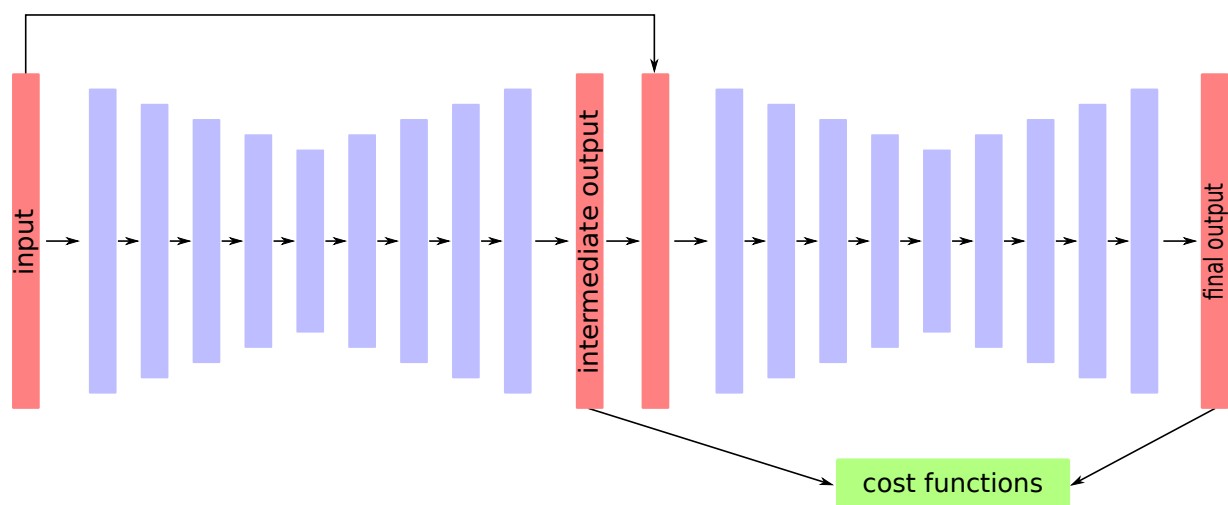

**Figure 2.** DINCAE with a refinement step composed essentially by two sequential autoencoders coupled such that the second autoencoder uses the output of the first and the input data.

The altimetry data was analysed on a 0.25 degree resolution grid covering the area from $7^oW$ and $37^oE$ and $30^oN$ to $46^oN$. For this neural network implementation, the input size is $177 \times 69$. The resolution is progressively reduced to $89 \times 35$, $45 \times 18$ and $23 \times 9$ by convolutional layers followed by max pooling layers with 32, 64 and 96 convolutional filters respectively. Skip connections are implemented after the 2nd convolutional layer onwards.

During training, Gaussian noise with a standard deviation $\sigma_{\text{pos}}$ is added to the position of the measurements and every track of the current date has a certain probability $p_{\text{drop}}$ to be withheld from the input of the neural networtk. The loss function is computed only on tracks from the current date. This helps the neural network to learn how to spread the information spatially. The neural network is optimized during 1000 epochs and the intermediate results are saved every 25 epochs.

The altimetry test case illustrates the results for a non-gridded dataset. Sea surface altimetry is usually gridded with a method like optimal interpolation or variational analysis. The latter can also be seen as a special case of optimal interpolation. For the autoencoder, the following fields are used as inputs:

– longitude, latitude of the measurement

– day of the year (sinus and cosinus) of the measurement multiplied by $2\pi/T_y$ where $T_y$ is 365.25 (the length of a year in days).





- all data within a given centered time window of length $\Delta t_{\text{win}}$. For instance if the time window of length $\Delta t_{\text{win}}$ is 9, the data from the current day are used as well as the tracks from the 4 previous days and the 4 following.

As in Barth et al. (2020), instead of using the observations directly, the observations are divided by their respective error variance and the inverse of the error variance is used as input. After this scaling, it is thus natural to consider missing data as

zero values as a zero corresponds to a data point with an "infinitely" large error.

The training is done using mini-batches of size $n_{\text{batch}}$. The weights and biases in the neural network are updated using the gradient of the loss function evaluated at a batch of $n_{\text{batch}}$ time instances (chosen at random). Evaluating the gradient using a subset of the training data chosen at random introduces some stochastic fluctuation allowing the optimization procedure to escape a local minima.

All numerical experiments used the Adam optimizer (Kingma and Ba, 2014) with the standard parameter for the exponential decay rate for the first moment $\beta_1 = 0.9$, and for the second moment $\beta_1 = 0.999$, and a regularization parameter $\epsilon = 10^{-8}$. The learning rate $\alpha_n$ is computed for every $n$-th epoch as follows:

$$\alpha_n = \alpha_0 \, 2^{-\gamma_{\text{decay}} n}$$

where $\alpha_0$ is the initial learning rate and $\gamma_{\text{decay}}$ controls the exponential decay of the learning rate: every $1/\gamma_{\text{decay}}$ epochs the learning rates is halved. If $\gamma_{\text{decay}}$ is zero, then the learning rate is kept constant.

The batch size includes 32 time instances (all hyper-parameters are determined via Baysian optimization as explained in the following). The learning rate for the Adam optimizer is 0.000579728. The L2 regularization on the weights has been set to a $\beta$ value of $10^{-4}$. The upsampling method can either be by nearest neighbor or bilinear interpolation. In our tests, the nearest neighbor provided the lowest RMS error relative to the development dataset. The absolute value of the gradients is clipped to 5 in order to stabilize the training. Satellite tracks from $\Delta t_{\text{win}} = 27$ time days (centred around the target time) are used to derive

gridded altimetry data.

The hyper-parameters mentioned previously of the neural network have been determined by Bayesian optimization (Mockus et al., 1978; Jones et al., 1998; Snoek et al., 2012) by minimizing the RMS error relative to the development dataset. The so-called expected improvement acquisition function (Mockus et al., 1978) is used to determine which sequence of parameter values is to be evaluated. Bayesian optimization as implemented by the Python package scikit-optimize (Head et al., 2020) was

used in these tests.

## 5 Results and discussion

### 5.1 Gridded data

The new type of skip connection was first tested with the AVHRR Test case from the Ligurian Sea (Barth et al., 2020). The previous best result (in terms of RMS) was $0.3835^o$C using the cat-skip connection. With the new approach this RMS error is





reduced to 0.3604$^{o}$C. The new type of skip connection makes the neural network more similar to residual networks which have been shown to be highly successful for image recognition tasks and allow to train deep networks more easily (He et al., 2015).

In table 2 we present the test case for the Adriatic Sea with and without the skip connections and using the multivariate reconstruction.

**Table 2.** RMS errors (in degree Celsius) relative to the test dataset for different configurations (chlor_a: Chlorophyll a, wind_speed: the wind speed, uwnd: zonal wind component, vwnd: meridional wind component).

| Auxilary parameters | cat skip-connections | sum skip-connections | sum skip-connections and refinemenet |
|---|---|---|---|
| none | 0.6559 | 0.5991 | 0.5464 |
| chlor_a | 0.6396 | 0.5949 | 0.5442 |
| chlor_a, wind_speed | 0.6460 | 0.5807 | 0.5397 |
| chlor_a, wind_speed, uwnd, vwnd | 0.6621 | 0.5720 | 0.5378 |

5  When reconstructing sea surface temperature time series, it is often the case that for some days only very few data points are available. Figure 3 illustrates such a case where a quite clear image was almost entirely masked as missing. DINCAE essentially produces an average SST (still using previous and next time frames) for the time of the year and a realistic spatial distribution of the expected error. The few pixels that are available have a relatively low error as expected, but the overall error structure looks quite realistic as the expected error increases significantly in the coastal areas (where the variability is higher

10  and where the original satellite data expected to be noisier). The reconstructed image matches the original image large-scale patterns relatively well, but as expected some small-scale structures are not reconstructed by the neural network.



## Date: 2003-03-21



**Figure 3.** Panel (a) the original MODIS SST, (b) MODIS SST with additional clouds for cross-validation, (c) the DINCAE reconstruction using the data from panel (b), (d) the expected error standard deviation of the DINCAE reconstruction. All panels are in degrees Celsius.

The detection of cloud pixels in the MODIS dataset is generally good, but Figure 4) for 2003-09-17 shows an example where some pixels were characterised as valid sea points while probably they are (at least partially) obscured by clouds, resulting in an unrealistic low sea surface temperature. For most analysis techniques derived from optimal interpolation, outliers like undetected clouds typically degrade the analysed field in the vicinity of the spurious observations. The outlier also produced



an artefact in the output of DINCAE, but it is interesting to note that in this case, the artefact did not spread spatially and the
associated expected error has some elevated values indicating a potential issue at the location.

## Date: 2003-09-17

### (a) Original data
### (b) With added clouds (3600)
### (c) DINCAE reconstruction
### (d) Exp. error std. dev.

**Figure 4.** Panel (a) the original MODIS SST, (b) MODIS SST with additional clouds for cross-validation, (c) the DINCAE reconstruction
using the data from panel (b), (d) the expected error standard deviation of the DINCAE reconstruction. All panels are in degrees Celsius.

A problem with techniques like optimal interpolation, variational analysis and to some degree also DINEOF, is that the
reconstruction smoothes-out some small-scale features present in the initial data. For optimal interpolation and variational
analysis, this smoothing is explicitly induced by using a specific correlation length. In EOF-based methods, this is related to



the truncation of the EOFs series. In DINCAE, the input data is also compressed by a series of convolution and max-pooling layers, and some smoothing is also expected, as in figure 4. Figure 5 shows an example where the initial data has almost no clouds and only few clouds are added for validation. The reconstructed field retains the filament and other small-scale structures.



**Figure 5.** Panel (a) the original MODIS SST, (b) MODIS SST with additional clouds for cross-validation, (c) the DINCAE reconstruction using the data from panel (b), (d) the expected error standard deviation of the DINCAE reconstruction. All panels are in degrees Celsius.





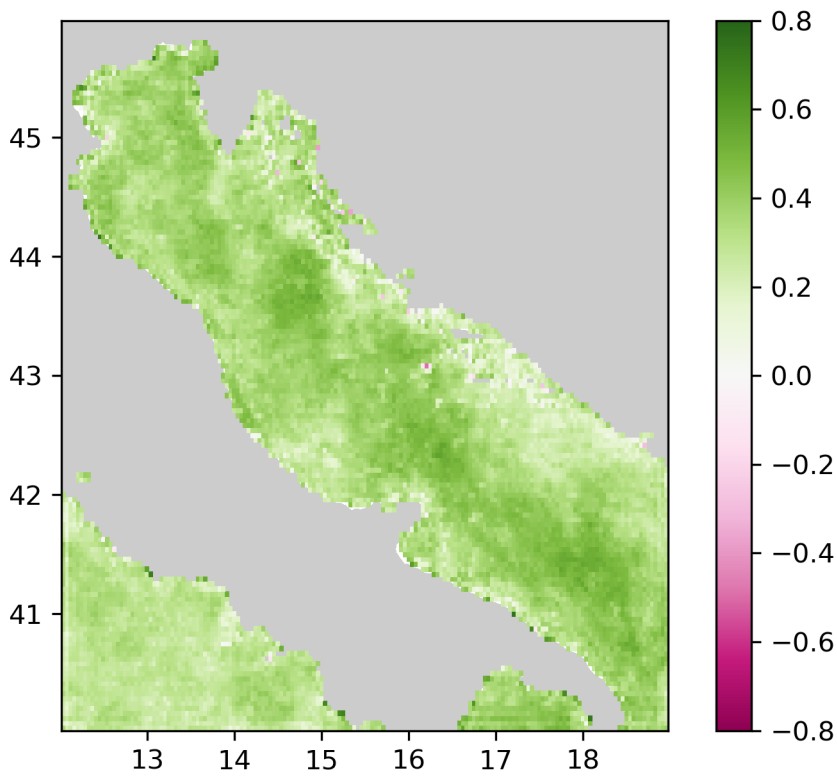

**Figure 6.** Mean square error skill score of the monovariate reconstruction corresponding to DINCAE 1.0 and the multivariate case (considering all variables) and with an additional refinement step.

To assess the improvement spatially, the mean square skill score $S$ is computed (Figure 6) for every grid cell.

$$S = 1 - \frac{\text{MSE}}{\text{MSE}_{\text{ref}}} \tag{8}$$

where $\text{MSE}_{\text{ref}}$ is the mean square error of the monovariate reconstruction corresponding to DINCAE 1.0 relative to the validation dataset (per grid cell and averaging over time) and MSE is the mean square error of the multivariate case (considering all variables) and with an additional refinement step. The improvement is spatially quite consistent. The mean square error is mostly reduced in the northern and central parts of the Adriatic. Only on some isolated grid cells a degradation is observed. The skill score reflects the combined improvement due to the 3 changes implemented in this version: updated skip-connections, refined step and multivariate reconstruction.



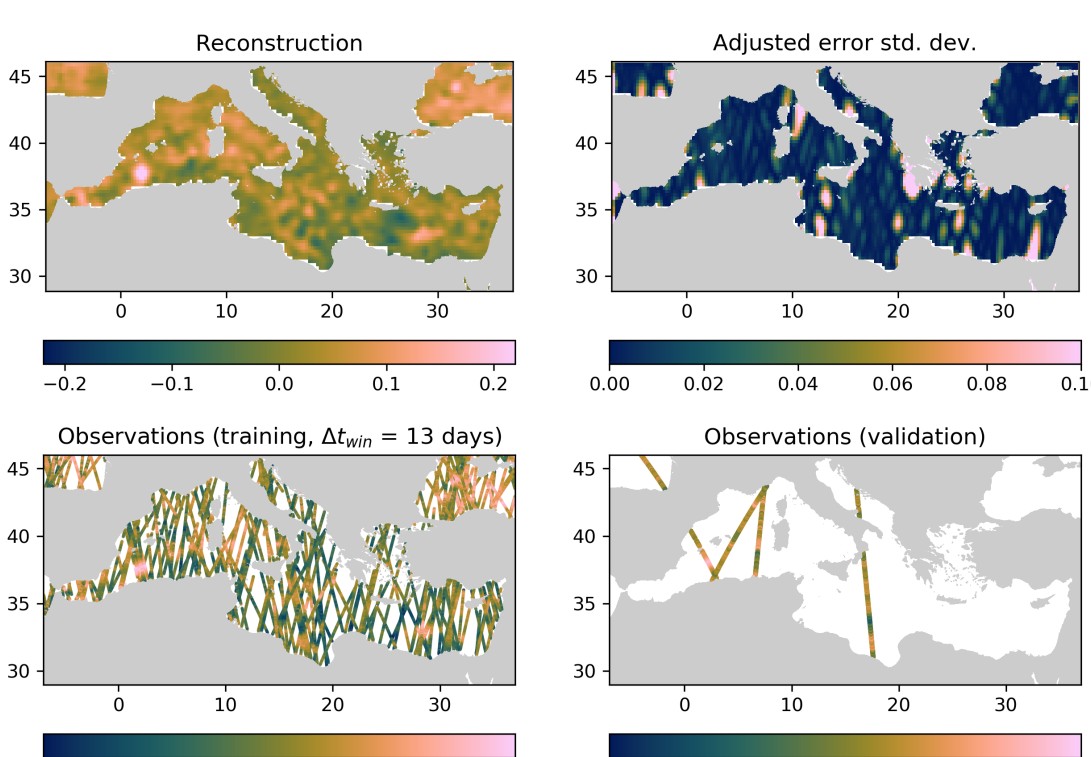

**Figure 7.** Panel (a) Reconstructed SLA by DIVAnd, (b) Expected error standard deviation by DIVAnd (without adjustment), (c) Data used during training (partial), (d) Independent data for validation withheld during analysis. All panels are in meters.





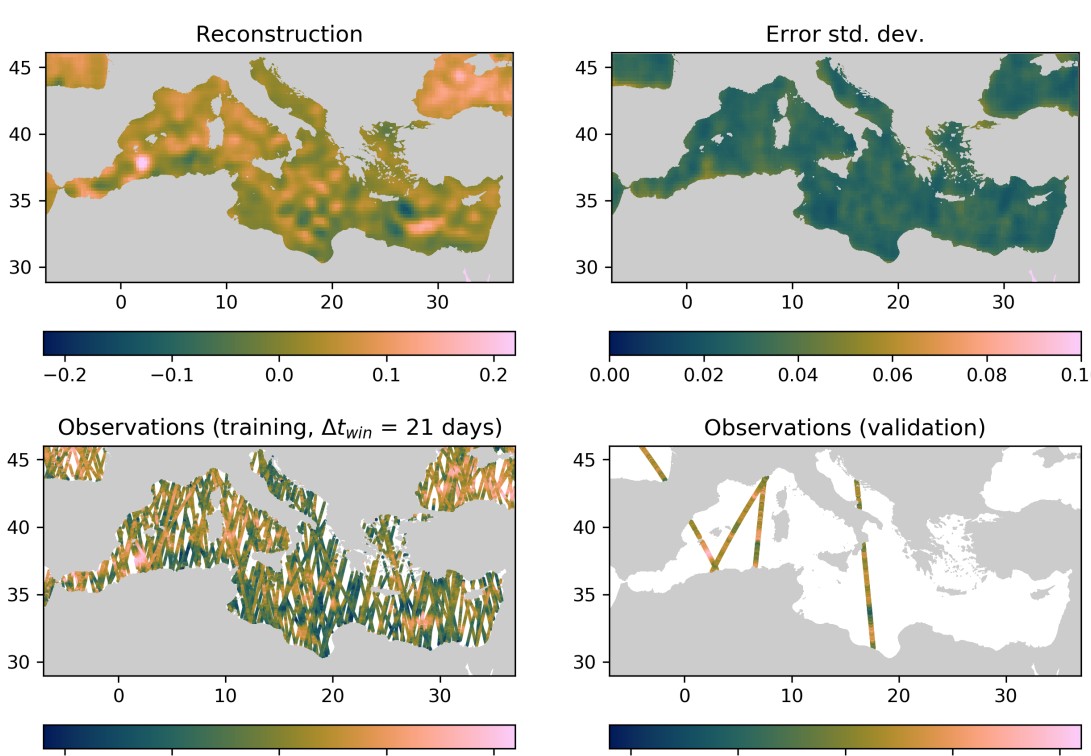

**Figure 8.** Panel (a) Reconstructed SLA by DINCAE, (b) Expected error standard deviation by DINCAE, (c) Data used during training (partial), (d) Independent data for validation withheld during analysis. All panels are in meters.



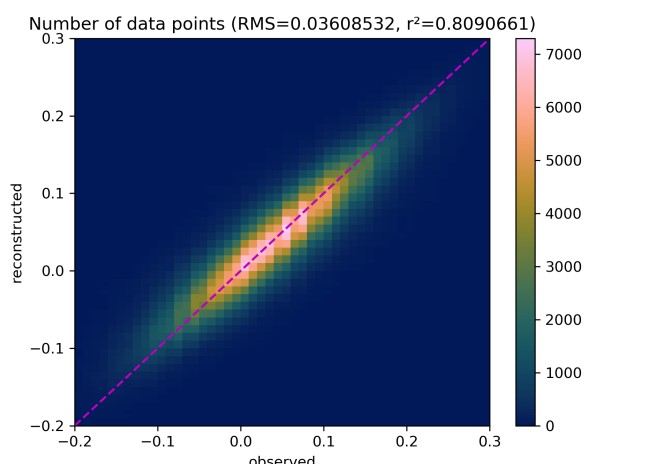
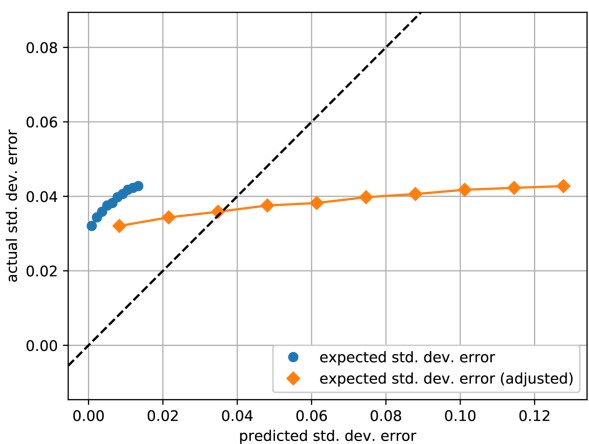

**Figure 9.** Left panel: altimetry observation from the test data versus the reconstructed values from DIVAnd (using only the training data). Right panel: expected standard deviation of the reconstruction (before and after adjustment) relave to the actual standard deviation of the reconstructed missfit.

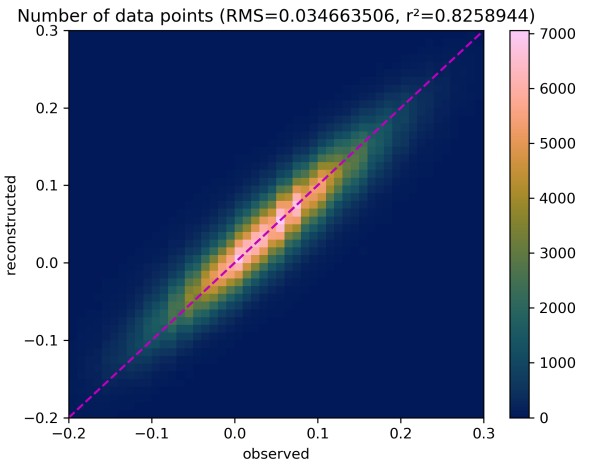
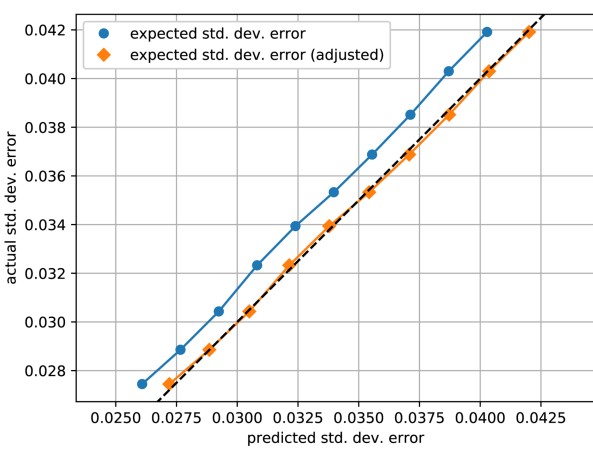

**Figure 10.** Left panel: altimetry observation from the test data versus the reconstructed values from DINCAE with SST as auxiliary parameter (using only the training data). Right panel: expected standard deviation of the reconstruction (before and after adjustment) relave to the actual standard deviation of the reconstructed missfit.

## 5.2 Non-gridded data

The altimetry data is first gridded by the tool DIVAnd (Barth et al., 2014). The main parameters here are the spatial correlation length (in km), the temporal correlation scale (days), the error variance of the observations (normalized by the background error variance) and the duration of the time window $\Delta t_{\mathrm{win}}$ determining which observations are used to compute the reconstruction





at the center of the time window.

All parameters of DIVAnd are also optimized using Bayesian minimization with expected improvement as acquisition function by minimizing the RMS error relative to the developpement datasets.

The best DIVAnd result is obtained with a horizontal correlation length of 74.8 km, a temporal correlation length of 5.5 days, a time window of 13 days and a normalized error variance of the observations of 20.5. An example reconstruction for the date 2017-06-07 is illustrated in Figure 7. The parameters are determined by Bayesian optimization minimizing the error relative to the developpement dataset. The RMS error of the analysis for these parameters is 3.61 cm relative to the independent test

dataset (Figure 9).

The best performing neural network had a RMS error of 3.58 cm which is only slightly better than results of DIVAnd (3.60 cm). When using the Mediterranean sea-surface temperature as a co-variables we obtained a RMS error relative to the test dataset of 3.47 cm resulting in a clearer advantage of the neural network approach (Figure 10). The eddy field of this data is quite similar to the one obtained from DIVAnd, but the anomalies of the structure are more pronounced in the DINCAE

reconstruction (Figure 8).

DINCAE and DIVAnd provide a field with the estimated expected error. For DIVAnd we used the so-called "clever poorman's method" as described in Beckers et al. (2014) and the background error variance is estimated by fitting an empirical covariance based on a random pair of points binned by their distance (Thiebaux, 1986; Troupin et al., 2012). The estimated

error standard deviation is later adjusted by a factor to account for uncertainties in estimating the background error variance.

We made 10 categories of pixels based on the expected standard deviation error, evenly distributed between the 10th and 90th percentile of the expected standard deviation error. For every category we computed the actual RMS relative to the test dataset. Ideally this should correspond to the estimated expected error of the reconstruction (including the observational error).

A global adjustment factor is also applied so that the average RMS error matches the mean expected error standard deviation which is represented in the left panels of Figures 9 and 10. The main advantage of DINCAE relative to DIVAnd is the improved estimate of the error variance of the results.

In summary, the accuracy of the DINCAE reconstruction is slightly better than the accuracy of the DIVAnd analysis. How-

ever, the main improvement of the DINCAE approach here is that the expected error variance of the analysis is much more reliable than the expected error variance of DIVAnd.





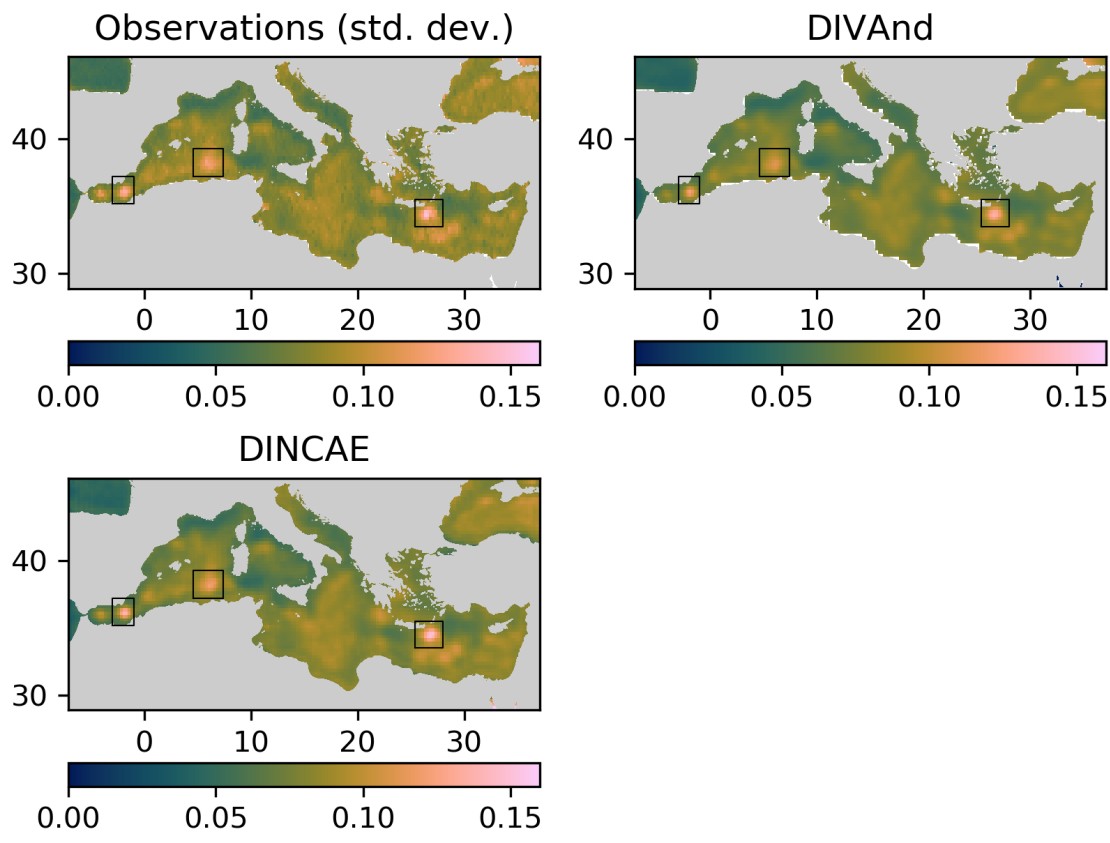

**Figure 11.** Standard deviation of the sea-level anomaly for the DIVAnd method and DINCAE (including SST as auxiliary parameter)

Figure 11 shows the standard deviation of the sea-level anomaly computed over the whole time period. From this figure, three areas in particular stand out corresponding (from East to West) to the East Alboran Gyre, regions of the Algerian current and the Ierapetra Anticyclone (annotated with the black rectangle in Figure 11). The maximum standard deviation (related to the surface transport variability) for these 3 areas is shown in Table 3. The standard deviation is also computed from the satellite altimetry data considering all satellite observations falling within a given grid cell (excluding coastal grid cells with less than 10 observations). The standard deviation of the DINCAE reconstruction is in all 3 regions higher than the standard deviation for DIVAnd despite that the DINCAE reconstruction has a lower RMS error than DIVAnd. Also the standard deviation of DINCAE is in general closer to the observed standard deviation.





**Table 3.** Maximum standard deviation in three selected areas

|  | Obs. std. dev. | DIVAnd std. dev. | DINCAE std. dev. |
|---|---|---|---|
| East Alboran Gyre | 0.136 | 0.123 | 0.141 |
| regions of the Alboran current | 0.125 | 0.112 | 0.121 |
| Ierapetra Anticyclone | 0.153 | 0.138 | 0.151 |

## 6 Conclusions

In this paper, we discussed improvements of the previous described DINCAE method. The code has been extended to handle multi-variate reconstructions which were also described in Han et al. (2020). We also found that multivariate reconstruction can improve the reconstruction, but the largest improvement was obtained by changing the structure of the neural network by using

a newly implemented different type of skip connection and refinement pass. Interestingly, this type bears some similarities to the hierarchical multigrid method for solving partial differential equations. The handling of different types of satellite data was also improved. While most ocean satellite observations are gridded data (like sea surface temperature, ocean color, sea surface salinity, sea ice concentration), some parameters can only be inferred by nadir looking satellites along tracks. For such non-gridded datasets the first input layer is extended to handle such arbitrarily location input data. We were able to show that

the DINCAE method applied with altimetry data produces better reconstructions, but the main advantages are the significantly improved estimates of the expected reconstruction error variance. In this case, the DINCAE method compares to the DIVAnd method (which is similar to optimal interpolation) favorably in terms of reliability of the expected error variance, accuracy of the reconstruction relative to the test dataset and realism of the temporal standard deviation of the reconstruction assessed from the standard deviation of the observations.

*Code and data availability.* The source code is released as open source under the terms of the GNU General Public Licence v3 (or, at your option, any later version) and available at the address https://github.com/gher-ulg/DINCAE.jl in the branch dev (doi: 10.5281/zenodo.5575067). The sea surface temperature (MODIS Terra Level 3 SST Thermal IR Daily 4km Nighttime v2014.0, doi:10.5067/MODST-1D4N4) is available by PO.DAAC (https://podaac.jpl.nasa.gov/, JPL, NASA, US), Wind speed (Cross-Calibrated Multi-Platform (CCMP) gridded surface vector winds) is available from Remote Sensing Systems (http://www.remss.com/measurements/ccmp/). Chlorophyll-a from

Ocean Biology Processing Group, NASA (US) can be accessed at. https://oceancolor.gsfc.nasa.gov/data/overview/. Altimetry data is made available by the Copernicus Marine Environment Monitoring Service (CMEMS, https://marine.copernicus.eu/)
(dataset SEALEVEL_EUR_PHY_L3_REP_OBSERVATIONS_008_061). The L4 gridded SST over the Mediterranean is the NOAA Optimum Interpolation 1/4 Degree Daily Sea Surface Temperature (OISST) Analysis, Version 2 available at https://www.ncei.noaa.gov/metadata/geoportal/rest/metadata/item/gov.noaa.ncdc:C00844/html (doi:10.7289/V5SQ8XB5).



*Author contributions.* A.B. designed and implemented the neural network. A.B., A.A.A., C.T. and J.M.B. contributed to the planning and discussions and to the writing of the manuscript.

*Competing interests.* The authors have no competing interests

*Acknowledgements.* The F.R.S.-FNRS (Fonds de la Recherche Scientifique de Belgique) is acknowledged for funding the position of Alexander Barth. This research was partly performed with funding from the Belgian Science Policy Office (BELSPO) STEREO III program in the framework of the MULTI-SYNC project (contract SR/00/359). Computational resources have been provided in part by the Consortium des Équipements de Calcul Intensif (CÉCI), funded by the F.R.S.-FNRS under Grant No. 2.5020.11 and by the Walloon Region. The authors wishes also to thanks to Julia community and in particular Deniz Yuret from the Koç University (Istanbul, Turkey) for the Knet.jl package and Tim Besard (Julia Computing, Massachusetts, United States) for the CUDA.jl package as well as the developers of the python library scikit-optimize.



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
