# Peer review of "DINCAE 2: multivariate convolutional neural network with error estimates to reconstruct sea surface temperature satellite and altimetry observations"

_Geoscientific Model Development, 2021_

## Author Comment (AC1)

**Reply to reviewer 2: DINCAE 1: multivariate convolutional neural network with error estimates to reconstruct sea surface temperature satellite and altimetry observations**

We thank the reviewer for the careful reading of the manuscript and the constructive criticisms. We copied the reviewer's comments below and our responses are in bold.

The paper presents an update on the previous version of DINCAE, a convolutional autoencoder method for in-painting of sparse satellite data. DINACE2 presents some improvements over the previous version, most notably in performance (vs DINCAE1), speed (presumably due to being rewritten to Julia from Python) and an option to treat ungridded data like satellite altimetry observations. It also introduces an extra refinement step in the cost function to increase its depth, and an intermediate loss term is included in the total loss to compensate for the vanishing gradients of the deep network. When treating sparse data, the error variance estimation of DINCAE2 is more reliable than that from variational interpolation method DIVAnd. The results are solid, the paper is well written, the figures are clear. I recommend publication after minor revision. I do have some comments which might be worth discussing further.

**Specific comments**

- page 4, eq4: In the comment to equation 4, the authors state that CAE refinement leads to a deeper network and thus to potential worsening of the vanishing gradient problem. They attempt to mitigate this by including intermediate loss term into the total loss function. They state that by doing so, the vanishing gradient problem is reduced. Can the authors perhaps illustrate more clearly that this is indeed the case? Is there a way to say a bit more about this, so that the reader does not have to take the author's word for this?

  **We added the following to the manuscript to clarify this question about the vanishing gradient:**

  **With a refinement step, the neural network becomes essentially twice as deep and the number of parameters (approximately) doubles. The increased depth would make it prone to the vanishing gradient problem. However, by including the intermediate results in the cost function this problem is reduced. In fact, the information from the observations is injected during back-propagation by the loss function. Due to the refinement step and the loss function which also depends on the intermediate result, the information from the observation is injected at the last layer and at the middle layer of the combined neural network (Szegedy et al., 2015). The relationship between the first layers of the neural network and the cost function is therefore more direct, which helps in the training of these first layers.**

- Also, wouldn't an arbitrary number of refinements further exacerbate the vanishing gradient problem? Which term would be dominant in this case – adding further refinement steps versus including further intermediate losses to the total loss?

  **It is important to note that for every refinement step an additional term must be added to the total loss. The number of refinement steps and terms in the loss function cannot be varied independently. In simple terms, the vanishing gradient problem can be expressed by the distance (counted as the minimal number of intermediate layers) between each layer and the loss function. For regular deep neural networks (without refinement), this distance would increase in average with the number of layers. However, this distance would be kept constant if additional refinement steps were added. In practice, adding**

a refinement step would double the amount of required GPU memory (two additional refinements, triple the amount of GPU memory,...). Regarding in-painting applications, we are only aware of applications with a single additional refinement step (Liu et al., 2019) while for image classification the loss function of the Inception network provides class labels at three different steps of the network, which is conceptually similar to the refinement approach used in our work (Szegedy et al., 2015).

- Page5: when handling missing data there is an interpretation throughout the text that setting the missing value to zero corresponds to an infinitely large error. This is undoubtedly true for variables which are normalized by their variance, as those in this paper. However there are a number of other scalings where variables are not normalized by their variance. In these cases, it seems to me, the authors interpretation is not the most appropriate. I would propose an independent interpretation that setting the missing values to zero simply numerically means that there will be no back-propagation of error from those missing data – thus the training can continue without any impact from the missing data. This interpretation does not have anything with the specific variable normalization at hand.

  **For the output value of the network, one can indeed ignore the missing values e.g. on land grid cells (this is also done here) and their associated value (0 or anything else) would not be used to update the network when computing the gradients via back-propagation. However, the more difficult aspect is how to treat missing values in the *input*. If the inputs are normalized, then a zero input value could either be the average or a missing value. To disambiguate between the two cases, we think it is important to provide an additional field which could either be a binary mask or the inverse of the error variance (both would be equivalent if the error variance of present data is assumed constant, as it is the case here). Other scales might be possible leading to a different interpretation and handling of missing value, but we found that the present one is the most natural from a data assimilation perspective when using Gaussian distributed errors (as discussed in Barth et al. (2021), equation 1 and 2). Once the scaling is chosen, the interpretation of missing values follows from the scaling. We clarified in the manuscript that our interpretation is specific to the chosen scaling.**

- A cosmetic remark. The hyperparameters were tuned using Bayesian optimization, which seems adequate. Let's say hyperparameter optimization gives you an optimal network. Separate instances of training this same optimal network (with the same fixed optimal hyperparameters) would provide separately trained versions of this same network. We can use these set of the same network to create a set of predictions. What is the error variance of this set of predictions? I would expect that this error variance is on the order of a 5

  **We agree with this comment and reduced the number of decimals in Table 2. The revised table now has only 2 decimals. This way we could also more easily add more information in this table as asked by the other reviewer.**

**References**

Barth, A., Troupin, C., Reyes, E., Alvera-Azcárate, A., Beckers, J.-M., and Tintoré, J.: Variational interpolation of high-frequency radar surface currents using DIVAnd, Ocean Dynamics, 71, 293–308, doi: 10.1007/s10236-020-01432-x, 2021.

Liu, H., Jiang, B., Xiao, Y., and Yang, C.: Coherent Semantic Attention for Image Inpainting, in: 2019 IEEE/CVF International Conference on Computer Vision (ICCV), pp. 4169–4178, doi:10.1109/ICCV.2019.00427, URL `http://openaccess.thecvf.com/content_ICCV_2019/html/Liu_Coherent_Semantic_Attention_for_Image_Inpainting_ICCV_2019_paper.html`, 2019.

Szegedy, C., Liu, W., Jia, Y., Sermanet, P., Reed, S., Anguelov, D., Erhan, D., Vanhoucke, V., and Rabinovich, A.: Going Deeper with Convolutions, in: Computer Vision and Pattern Recognition (CVPR), URL `http://arxiv.org/abs/1409.4842`, 2015.

---

## Author Comment (AC2)

**Reply to reviewer 2: DINCAE 2: multivariate convolutional neural network with error estimates to reconstruct sea surface temperature satellite and altimetry observations**

**We thank the reviewer for the in-depth look and the many insightful comments. manuscript and the constructive criticisms. We copied the reviewer's comments below and our responses are in bold.**

**General comments**

The paper by Barth et al. (2021) presents a modification of the DINCAE method for reconstructing missing data from satellite sources. The main advantages of this updated procedure lie on the ability to handle multivariate and non-gridded data, on the superior performance given by an innovative form of the skip connection and the addition of a refinement pass. Moreover, in order to prove the improvements with respect to the previous version, some applications are shown. The main result of the paper concerns the accuracy of the reconstructed error variance used in the refinement of the cost function when dealing with non-gridded datasets. The computational speed improvements are probably due to the change in the programming language. Overall, the paper shows interesting results concerning the accuracy of reconstructed sparse data but more attention to the scientific rigor and the clarity of concepts and figures is needed. I recommend publication with minor revisions, although I suggest paying careful attention in resolving them.

**Specific comments**

- I think that you use the words "variance" and "standard deviation" in an equivalent manner. The standard deviation is the square root of the variance that you call both $\sigma$ throughout the paper (see for example in Section 2.2 or Section 4). Also I think that sometimes the same variable is also mistaken for the error associated with that variance. Please, revise their definition and be consistent with the name of the variables.

  **Of course we agree with the definition of the reviewer ($\sigma$: standard deviation and $\sigma^2$: variance). We apologize for the confusion in the paper and we're thankful for the reviewer for having spotted this issue. "Variance $\sigma$" should have been written as "Variance $\sigma^2$" (and it is not a misguided attempt to redefine the symbol $\sigma$).**

- Pg. 4: In the definition of the loss function (Eq. 3) a term is missing, which I think is negligible for your purpose but needs to be there (or, at least, you should mention that you are neglecting it). Also it is a good practice to specify what represents each variable before or after every equation.

  **We added the constant term $2\log(\sqrt{2\pi})$ in this equation with the note:**

  **The third term on the right hand-side is constant and can be neglected as it does not influence the gradient of the cost function.**

  **In the revised manuscript we also now define $i$, $j$ and $N$ (whose definitions were indeed missing in the original version).**

- Pg. 5: Equation (5) is incorrect. It should be:

$$g(x,y) = \int w(x - x', y - y')f(x', y')dx'dy'.$$

We thank the review for pointing this out. This is indeed a typo and inconsistent with the follow-up equations. We corrected the revised version of the manuscript. Luckily this typo has no bearing on the results of the manuscript.

- In Section 3.2 you explain that in preliminary studies you found that including the data from adjacent seas can help a neural network to better generalize and to prevent overfitting. Can you cite some valid reference or say a little bit more about that? Do you have some results that you can show or are you able to give a reasonable explanation on this topic?

  While not specific to machine learning in oceanography, the work from Gong et al. (2019) presents the benefit for data diversity while training machine learning models. This work shows that diversity in the datasets in fact prevents overfitting and aids in generalization. We have added this to the manuscript.

- I strongly suggest to spend more words analyzing the results of Table 2 and all the figures thereafter, including (average, max or min) values of the calculated errors with respect to the original data in the region.

  We agree and we have expanded the discussion of Table 2 and the figures. In particular we added in Table 2 for every experiment the 10% and 90% percentile of the absolute value of the difference between the reconstructed data and the cross-validation data to provide a typical range of the error. We prefer to use percentiles as they are more robust to statistical noise than the minimum and maximum values as only one grid cell has an error equal to the minimum (or maximum) error. These statistics were also added when discussing the related Figures 3 to 5. For Figure 5, we also quantified the degree of smoothing by computing the RMS difference of the reconstructed data and the input data.

- The last sentences of Sec. 5.2 (lines 6-8 of page 21) seem to state that even if the SD of the DINCAE reconstruction is higher, the RMS is lower. I do not see the contradiction in that, in fact the reconstructed SD from the DINCAE method is on average closer to the observed one than the DIVAnd's one. The point is not that the SD is higher or lower but simply more or less accurate with respect to the one derived from the DIVAnd method, and it would be nice to show a quantity that describes this improvement. Maybe you could further describe this improvement showing a whole map of the errors over the whole time period.

  What we had in mind when we wrote this, is the equation linking the RMS error and the standard deviation of the reconstruction ($\sigma_r$) as shown by e.g. Taylor (2001):

  $$\mathbf{RMS}^2 = \mathbf{b}^2 + \sigma_r^2 + \sigma_o^2 - 2\sigma_r\sigma_o r$$

  Where $b$ is the bias, $\sigma_o$ the standard deviation of the observation and $r$ is the correlation coefficient. If the correlation is low, then a comparison of reconstruction methods based on the RMS error would favor reconstructions methods with a low standard deviation $\sigma_r^2$. Only if the correlation is high, the last term of this equation can offset the contribution of $\sigma_r^2$ to the RMS error. We see this discussion analogous to the double penalty error (Shapiro et al. 2021) in numerical ocean modeling where higher resolution models often have a larger RMS error than coarse resolution models when some of the circulation features are present but misplaced (therefore low correlation $r$ for the high-resolution model) in the high-resolution model and when they are absent in the coarse resolution

model. In this case, the standard deviation of the coarse resolution model is smaller than the standard deviation of the high resolution one, yet the RMS error is typically smaller for the coarse resolution model (due to the double penalty effect). For us the main quantity describing the accuracy of the reconstruction is the RMS error (and to a lesser degree the correlation $r$ and the graphical validation of figures 9 and 10). Using the RMS metric, the DINCAE result and DIVAnd results are very close and for us the main advantage of DINCAE result is the more accurate estimation of the error variance of the reconstruction. As an additional metric we used the skill score proposed by Taylor (2001) defined as:

$$S = \frac{4(1 + R)}{(\frac{\sigma_r}{\sigma_o} + \frac{\sigma_o}{\sigma_r})^2(1 + R_0)}$$

where $R_0$ is the maximum attainable correlation. Even assuming that perfect correlation is possible, both techniques have very good skill scores (S = 0.952 for DINCAE and S = 0.940 for DIVAnd) as the maximum value of S is 1.

**Technical corrections**

- In general the paper presents a few errors (for example, several parentheses in citations are missing, singular/plural agreement in the sentences needs to be corrected and the adverbs position in several sentences is wrong). In particular, the word "data" is plural, please check the corresponding verb throughout the paper.

  **Indeed, we found 7 instances where we corrected the plural form of data.**

- Pg. 2, At the beginning of line 2 it should be "which" instead of "who".

  **Done, thanks!**

- Pg. 2, At line 13, since "deeper" is a comparative, the object of the comparison should be explained (deeper than what?). This is something that can be found several times in the paper.

  **We avoid the term "deeper network" in the revision. For instance, the mentioned paragraph now reads:**

  **We also investigate the influence of the depth of the neural networks in this work. It is known that neural networks are increasingly more difficult to train as their depth increases because of the well-known vanishing gradient problem (Hochreiter, 1998)**

- Pg. 3, It is not recommended to start a sentence with an equation (line 20-21). Also, it is a good practice to explain each variable when showing an equation, saying what X, f and l stands for in Eq (1). Same for equations (6) and (7).

  **We agree and have added a sentence before equation (1) explaining all the symbols in this equation. We also expanded the explanation for equation (6) and (7).**

- Pg. 3, At the end of Section 2.1 (line 28) I think "basis" should be "bias".

  **Yes, thanks a lot!**

- Pg. 3, Line 6: The word "error" appears twice.

  **Yes, this is corrected in the revised manuscript.**

- Pg. 3, Line 10: I think "expect" should be "except".

  **Yes, this is corrected in the revised manuscript.**

- Pg. 3, At the end of Section 2.2 (line 21) I think "gage" should be "gauge".

  **Yes, this is corrected in the revised manuscript.**

- Pg. 6 (line 26): The dataset SEALEVEL_EUR_PHY_L3_REP_OBSERVATIONS_008_061 changed its name in SEALEVEL_EUR_PHY_L3_MY_OBSERVATIONS_008_061 after some corrections. Can you please correct the name or declare the date in which you downloaded the data?

  **We updated the manuscript with the new name of the dataset and included the date of access. In fact, we redownloaded the dataset in October, 13th 2020 after we became aware of the issue CMEMS:11964 (`https://marine.copernicus.eu/newsflash/cmems11964-sealevel_eur_phy_l3_rep_observations_008_061-anomaly-regional-europe-along-track`). So the analysis in the paper is based on the updated datasets. The access date is now mentioned in the manuscript.**

- Pg. 10 (line 15): I would rather say "sine and cosine".

  **Thanks, this is changed in the revised manuscript.**

- Pg. 11 (line 16): Either you are missing a word when you say "as explained in the following..." or you should change it to "as described further on" or similar.

  **We agree and change it to "as described further on".**

- Pg. 17: In the caption of Figure 7 you state that the map for the expected error standard deviation by DIVAnd is without adjustment while the title of the panel suggests "Adjusted error std. dev.". Could you correct either one of them? Also in both captions of Fig.s 7 and 8 you mention the panels (a), (b), (c) and (d) but those letters do not appear in the images.

  **It is the adjusted expected error standard deviation by DIVAnd. We clarified the caption and text of the manuscript.**

- In Figures 9 and 10 there are no units. Also, the title of the left panels suggest "Number of data points" which you never explain nor comment in the text. Moreover, there is no explanation of the colors or the dashed lines in both plots.

  **We agree that we should have provided more information in these Figures.**

  **The left panel of these figures shows on the x and y axis the observed altimetry (and withheld during the analysis) and the corresponding reconstructed altimetry respectively. If we would have used a simple scatter plot, the amount of data to be plotted would make the graphics unreadable. Therefore the range of altimetry values from -20 cm to 30 cm**

was divided in 51 bins of 1 cm. The colors indicated then the number of data points within each bin. So the data in the left panel is indeed adimensional (but the axis are in meters).

The right panel is the actual and predicted standard deviation in meters.

The dashed line in all panels corresponds to the ideal line when reconstructed data correspond exactly to the observed altimetry and the predicted error corresponds exactly to the average actual error.

The manuscript has been updated accordingly and we thank the review for pointing this issue out.

**References**

Gong, Z., Zhong, P., and Hu, W.: Diversity in Machine Learning, IEEE Access, 7, 64 323–64 350, doi:10.1109/ACCESS.2019.2917620, 2019.

Hochreiter, S.: The Vanishing Gradient Problem During Learning Recurrent Neural Nets and Problem Solutions, International Journal of Uncertainty, Fuzziness and Knowledge-Based Systems, 06, 107–116, doi:10.1142/S0218488598000094, 1998.

Taylor, K. E.: Summarizing multiple aspects of model performance in a single diagram, Journal of Geophysical Research, 106, D7, 7183–7192, 2001.